# Health-Related Quality of Life in Long-Term Survivors of Non-Hodgkin Lymphoma: A French Population-Based Study

**DOI:** 10.3390/cancers17040711

**Published:** 2025-02-19

**Authors:** Stephane Kroudia Wasse, Emerline Assogba, Sebastien Orazio, Kueshivi Midodji Atsou, Cédric Rossi, Adrien Guilloteau, Sophie Gauthier, Stéphanie Girard, Jean-Marc Poncet, Gandhi Damaj, Xavier Troussard, Alain Monnereau, Sandrine Tienhan Dabakuyo-Yonli, Marc Maynadie

**Affiliations:** 1Registry of Hematological Malignancies of Côte d’Or, Dijon-Bourgogne University Hospital, 21000 Dijon, France; kmatsou@outlook.fr (K.M.A.); adrien.guilloteau@chu-dijon.fr (A.G.); sophie.gauthier@u-bourgogne.fr (S.G.); stephanie.boulanger@u-bourgogne.fr (S.G.); marc.maynadie@u-bourgogne.fr (M.M.); 2INSERM, UMR1231, Bourgogne Franche-Comté University, 21000 Dijon, France; eassogba@cgfl.fr; 3Breast and Gynecologic Cancer Registry of Côte d’Or, Georges François Leclerc Comprehensive Cancer Centre, 21000 Dijon, France; sdabakuyo@cgfl.fr; 4Registry of Hematological Malignancies of Gironde, Bergonié Institute, 33076 Bordeaux, France; s.orazio@bordeaux.unicancer.fr; 5Clinical Hematology Unit, Dijon Bourgogne University Hospital, 21000 Dijon, France; cedric.rossi@chu-dijon.fr; 6Registry of Hematological Malignancies of Caen, Caen University Hospital, 14000 Caen, France; poncet-m@chu-caen.fr (J.-M.P.); damaj-gl@chu-caen.fr (G.D.); troussard-x@chu-caen.fr (X.T.); 7Clinical Hematology Unit, Caen University Hospital, 14000 Caen, France; a.monnereau@bordeaux.unicancer.fr

**Keywords:** quality of life, population-based data, non-Hodgkin lymphoma

## Abstract

The authors assessed the Health-Related Quality of Life (HRQoL) among survivors of follicular lymphoma and diffuse large B-cell lymphoma 5–14 years post-diagnosis. This population-based study used data from three registries specialized in hematological malignancies in France. The data collected were compared to those of a normative general French cohort. Overall, 493 survivors completed the validated questionnaires and reported lower HRQoL compared to the general French population. Older age, underweight or obesity, comorbidities, anxiety, depression, and sexual satisfaction problems found to be the main factors associated with poorer HRQoL. Findings suggest taking account these factors for guiding supportive care and engaging in specific health promotion interventions.

## 1. Introduction

In France, hematological malignancies represented 12% of all new cases of cancer in 2018, and 63% of them were non-Hodgkin lymphomas (NHL) [1]. Five-year net survival over the period from 1989 to 2018 was increased more than 20% in follicular lymphoma (FL) and diffuse large B-cell lymphoma (DLBCL), with 86% for FL and 61% for DLBCL in 2018 [2,3]. Improved treatment efficacy likely explains these increases. Indeed, during this period, the addition of a monoclonal antibody against CD20 (mostly rituximab) to CHOP chemotherapy (cyclophosphamide, doxorubicin, vincristine, prednisone) became the gold-standard treatment for NHL. After relapse, intensive chemotherapy with autologous stem-cell transplant is recommended [4,5,6]. However, treatment-related toxicities, such as cytopenia, neuropathies, or sexual problems, impact general health, thus affecting quality of life (QoL) [7,8,9] The issue of health-related QoL (HRQoL) is thus becoming increasingly crucial, in view of the increasing survival rates for FL and DLBCL. In fact, minimizing sequelae and enhancing HRQoL is one of the foci of the French national 10-year cancer strategy for 2021–2030 [10].

HRQoL has been investigated in several NHL clinical trials, but the results may not be applicable to the overall real-life population of NHL patients, due to the strict selection criteria applied in such studies [11]. Using data from cancer registries with a wide range of sociodemographics could be a better approach, as they are more likely to be representative of NHL survivors in that geographical area [12]. A population-based study, carried out by our group, using data from a French registry, reported that the main factors associated with poor HRQoL among NHL survivors were demographic, psychological, and economic problems, but not clinical factors [13]. However, the number of patients in this survey was small, prompting us to perform wider HRQoL investigations based on population data collected by several registries, in order to obtain an exhaustive and representative record of all cancer cases occurring in the area covered, thus limiting selection bias.

In this context, using data from the three registries specialized in hematological malignancies in France, the aim of the present study was to compare the HRQoL of NHL survivors to that of the general French population and to identify factors impacting HRQoL in NHL survivors.

## 2. Materials and Methods

### 2.1. The Study Population

We conducted a population-based study using NHL survivors diagnosed between 2010 and 2018 and identified from three cancer registries specializing in hematological malignancies, namely in Côte d’Or, Gironde and Basse-Normandie, covering 5 French administrative Departments with a total of 3,693,712 inhabitants in 2023—5% of French population. The diagnosis of NHL was made according to the third edition of the International Classification of Diseases for Oncology codes (ICD-O-3) considering FL (9690/3, 9691/3, 9695/3, 9698/3, 9597/3) and DLBCL (9678/3, 9679/3, 9680/3, 9684/3, 9688/3, 9712/3, 9735/3, 9737/3, 9738/3) [14]. Vital status was updated using the National Register of Identification of Physical Persons to exclude patients who had died before 1 September 2023. Other forms of hematological malignancies at diagnosis and patients younger than 18 years of age were not included in the study. Adults with an unavailable recent residential address, who were unable to express their consent, or who refused to participate were excluded.

### 2.2. Data Collection

A cross-sectional design was conducted with data collection at a specific period. In September 2023, a letter containing the information leaflet of the study, as well as the study questionnaires and a prepaid return envelope, was sent to the home of all eligible patients. Patients were informed that non-participation would not have any consequence on their follow-up care. Patients who accepted to participate in the study were invited to return the completed questionnaires to the Côte d’Or registry, which centralized all the data collection. A reminder letter was sent for patients who had not answered after two months. Returned questionnaires did not provide information on patient identity; rather, a randomly created numerical identifier was used to blindly link questionnaires to the mailing list.

### 2.3. Measurements

Demographic variables (age, sex, date of birth) and NHL characteristics (ICDO3 codes, date of diagnosis, Ann Arbor stage) were collected from the registries databases.

The study outcome was HRQoL, assessed by the French-language version of the 12-item Short-Form health survey (SF-12), which is a generic questionnaire used for patients with cancer and normative populations. The SF-12 questionnaire generated eight scales and two summary components (physical and mental). All scale scores ranged from 0 to 100, with higher scores representing a better level of HRQoL. The SF-12 scores for respondent patients were compared to scores matched for gender and age from a normative French population (N = 25,931) [15]. Minimal clinically important differences were determined for each scale with the following rules: ≥2 points for role physical functioning; ≥3 points for physical functioning, social functioning, bodily pain, general health, vitality, mental health and component scales; ≥4 points for role emotional functioning [16].

Other sociodemographic, clinical and psychosocial characteristics were collected by questionnaires based on risk factors for HRQoL described in the literature [13].

Anxiety and depression were assessed by the validated Hospital Anxiety and Depression Scale (HADS) questionnaire, adapted in French in 1989 by Lepine et al. [17]. In this tool, total scores ranged from 0 to 21; a score greater than 11 indicates the presence of symptoms.

Social support was assessed by the validated 6-item Social Support Questionnaire (SSQ6) adapted in French by Rascle et al. in 2005 [18]. This questionnaire measured the availability of social support, on a scale of 0 to 54 and satisfaction with that support, on a scale from 6 to 36. Higher scores represent better perceived social support.

Social deprivation was determined by the validated French EPICES [19] questionnaire (“Assessment of Precariousness and Health Inequalities for Health Examination Centers”). The score ranges from 0 to 100; a score > 30 indicates social deprivation.

Sexuality was estimated by the EORTC Sexual Health Questionnaire (EORTC SHQ-22). It is composed of 2 multi-item scales to assess sexual satisfaction and sexual pain and 11 single-symptom items, with calculated scores ranging from 0 to 100. A higher score represents a higher level of symptoms or concerns [20].

Self-esteem was assessed using the Rosenberg questionnaire, validated in French [21]. It is composed of 10 items that generate a score between 10 and 40. A score < 31 indicates low self-esteem.

Other questions concerned income, educational level, alcohol consumption, smoking status, marital status, comorbidities, and weight and height computed into body mass index (BMI), which was then categorized into 4 classes according to the WHO recommendations [22], namely underweight: <18.5 kg/m^2^; normal weight: 18.5–25 kg/m^2^; pre-obesity: 25–30 kg/m^2^; and obesity: ≥30 kg/m^2^.

### 2.4. Statistical Analyses

Non-respondents and respondents were compared based on available characteristics from each cancer registry (age, time since diagnosis, gender, NHL sub-type, Ann Arbor stage, treatment, administrative department). For significantly different variables, an adjustment was performed including these variables in all multivariate analyses. All characteristics of respondents are described according to NHL sub-types. Data are described as mean and standard deviation (SD) or median and interquartile (IQR) for continuous variables, and as number and percentage for categorical variables. Mann–Whitney tests were used to compare continuous variables, and the chi square test was used for categorical variables. Student *t*-tests were used to compare each scale of HRQoL scores of respondent patients with those of the normative French population. *p*-values < 0.05 were considered statistically significant.

A multivariable regression model was used to assess statistical associations between each scale of HRQoL and sociodemographic (age as a continuous variable, gender, education level, income), lifestyle (smoking, alcohol consumption, sexual satisfaction problems), clinical (Ann Arbor stage, time since diagnosis as a continuous variable, comorbidities, BMI), and psychosocial factors (anxiety, depression, self-esteem, social satisfaction, economic deprivation). Correlations were tested between candidate variables at a statistically significant threshold of 0.05. The backward elimination method was applied, and a *p*-value < 0.157 was used to select predictor variables in the final multivariable models [23]. Mixed models were used in the multivariable model to account for the random effect of place of residence. A *p*-value < 0.01 was considered statistically significant for multivariable analyses.

All analyses were performed using SAS version 9.4 (SAS Institute Inc., Cary, NC, USA) and R version 4.3.1.

## 3. Results

In total, 1379 patients were eligible for the study and were contacted, of whom 493 returned the questionnaires, yielding a response rate of 36% (Figure 1). Table 1 shows the comparison between non-respondents and respondents. Among respondents, the median age was 67 years (IQR 58–75), and the median time since diagnosis was 8 years (IQR 6–10). Non-respondents were significantly different from respondents in terms of time since diagnosis (*p* = 0.007) and administrative residence (*p* < 0.001).

Among the characteristics of the respondents, DLBCL survivors were significantly different from FL survivors only in terms of R-CHOP chemotherapy (100% vs. 86%, *p* < 0.001; Table 2). Over half of respondents were men (56%), two-thirds were diagnosed with Ann Arbor stage III–IV (65%), more than half had no comorbidities (53%), income was between EUR 1500 and EUR 3000 per month in 43%, less than half had high self-esteem (48%), and the majority did not suffer from anxiety (82%) or depression (91%). Figure 2 indicates the sexual health parameters of respondents overall and according to NHL subtype. As shown in Figure 2a, for all respondents, the mean score for problems with sexual satisfaction was 37/100, while the mean score for communication problems about sexuality was 92/100. Respondents’ HRQoL scores were compared to those from the normative French population, matched for gender and age (Figure 3). All respondents had lower scores than the normative population on all subscales of HRQoL, except for bodily pain, regardless of NHL subtype (A. Overall: 71 vs. 66, *p* = 0.002; B. FL: 72 vs. 66, *p* = 0.01; C. DLBCL: 71 vs. 66, *p* = 0.04).

Table 3 reports the factors associated with the subscales of HRQoL by multivariable analysis using a mixed model. A one-year increase in age was associated with a significant decrease in general health (β = −0.2, *p* = 0.0001), physical scores (β = −0.3, *p* = 0.009), emotional scores (β = −0.2, *p* = 0.006), and bodily pain (β = −0.3, *p* = 0.004). A one-year increase in time since diagnosis was associated with an increase in social functioning (β = 1.2, *p* = 0.009). Men had better general health (β = 4.2, *p* = 0.01) and less bodily pain (β = 6.8, *p* = 0.007) than women. Higher income was associated with better HRQoL (*p* < 0.01). An increase in problems with sexual satisfaction was associated with poorer HRQoL (*p* < 0.01). Underweight and obesity were both associated with poorer physical functioning (*p* = 0.008). The presence of comorbidities, socioeconomic deprivation, anxiety, and depression were associated with poorer HRQoL (*p* < 0.01).

## 4. Discussion

In this population-based study, the HRQoL of NHL patients was evaluated extensively and compared to that of a French normative population, assessing demographic, clinical, socioeconomic, and psychological factors. We found that a longer time since diagnosis, male gender, and higher income were associated with better HRQoL in NHL patients. Conversely, older age, underweight or obesity, comorbidities, socioeconomic issues, anxiety, depression, and sexual satisfaction problems were associated with poorer HRQoL.

Comparison with the normative French population showed that the mean scores on all subscales of the SF-12 questionnaire were lower in the patient population, except for bodily pain. This trend has been reported in other studies and could be explained by the response-shift phenomenon, which is a change in the meaning of one’s self-evaluation of a target construct as a result of a change in the respondent’s internal standards of measurement, a change in the respondent’s values, or a redefinition of the target construct, being, in our study, the pain [13,14,15,16,17,18,19,20,21,22,23,24]. Indeed, patients are shown to redefine their internal standards in terms of symptoms according to their individual experience of the disease [25]. Moreover, clinically meaningful differences were observed between overall participants and the normative population for bodily pain, vitality, social functioning, and physical and emotional subscales [16,26].

Income is known to be a significant predictor of QoL, and people with higher income report better QoL. Indeed, a one-unit increase in income has been reported to increase scores on the mental health scale [27]. This was confirmed in our study, highlighting the issue of social inequalities. Targeted interventions are warranted in this domain to reinforce equity. It was also observed that men had better general health than women. Hjermstad et al. also found that women reported a lower overall health status [28], using the specific QLQC-30. These observations stress the need to take gender into account when health intervention strategies are planned for NHL survivors.

Increasing age was shown to be associated with poorer HRQoL, which is consistent with other reports [29]. Aging is characterized by physiological changes that can result in the occurrence of several diseases. Multimorbidity can result from a number of interrelated factors, such as the presence of one disease increasing the risk of developing another [30]. Aging and multimorbidity contribute to patient frailty and reduced HRQoL. However, in the present study, the multivariable analysis was adjusted for comorbidities, yet age was still found to be statistically significantly associated HRQoL, implying the need to take age into consideration in patient management.

Underweight or obesity and comorbidities were found to be associated with poorer HRQoL. Being overweight is a risk factor for many diseases, contributing to comorbidities such as coronary heart disease, heart failure, and stroke [31]. In cancer survivors, obesity has also been shown to be associated with a deterioration in HRQoL [32], and the well-established general recommendation of moderate to high levels of physical activity to maintain physical health also improves HRQoL in NHL survivors [33]. A longer time since diagnosis was associated with better social functioning. Strouse et al. [34] reported that whatever the history of treatment, with or without autologous hematopoietic cell transplantation, patients with aggressive lymphoma in remission at 3 and 6 years after diagnosis had a higher level of HRQoL [34]. Further studies using the cure model should assess whether cure is clinically meaningful. In the previous study by our group [13] that involved 157 NHL survivors, the presence of psychological factors, namely anxiety and depression, was found to be associated with poorer HRQoL. In the present study, with 493 NHL survivors, the strength of this finding is enhanced, strongly confirming the impact of psychological factors on poorer HRQoL. These results highlight the need for supportive psychological care on top of primary or specialty-related palliative care to improve HRQoL in NHL survivors [35,36]. Sexual problems are one of the most frequent consequences of cancer treatment and may persist for a long time, negatively impacting on HRQoL [8]. Here, the lack of communication about sexuality and the poorer HRQoL of overall survivors with sexual satisfaction problems was clearly highlighted. This indicates the need for caregivers to take communication about sexual concerns into account, in order to improve the HRQoL of NHL survivors.

One of the main strengths of this study is the use of multicenter data from three population-based cancer registries, covering a large population representative of NHL survivors, limiting selection bias. Other strengths are the long follow-up time since diagnosis, the adjustment for clinical factors in the multivariable analysis, the use of validated HRQoL questionnaires, and the comparison of HRQoL scores between overall survivors and the normative French population. This study provides reference HRQoL values among NHL survivors that could be used in other HRQoL studies, notably the real-life follow-up REALYSA [37] study (national study of lymphomas). Limitations of the study include, firstly, a response rate of 36%, which raises questions about the reasons why participants did not respond to the questionnaires. In elderly patients, cognitive problems, comorbidities, and poorer compliance with questionnaires may explain this rather low response rate [38], although it still yielded a highly informative and large population. Secondly, non-availability of a recent residential address contributed to the low rate of response. This is indeed one of the challenges of conducting studies from population-based data [12]. Moreover, the unavailability of R-CHOP chemotherapy duration and intensity was a limitation. Finally, the cross-sectional design of the study precludes any causal inferences between the factors identified and HRQoL. Further studies are required to investigate factors associated with HRQoL longitudinally.

## 5. Conclusions

At 5–14 years post-diagnosis, NHL survivors were shown to have poorer HRQoL than a French normative population, except in terms of bodily pain. Several key factors associated with poorer HRQoL in NHL survivors were identified. Moreover, our findings offer insights to physicians regarding the factors associated with HRQoL, which could be useful for guiding supportive care and engaging in specific health promotion interventions.

## Figures and Tables

**Figure 1 cancers-17-00711-f001:**
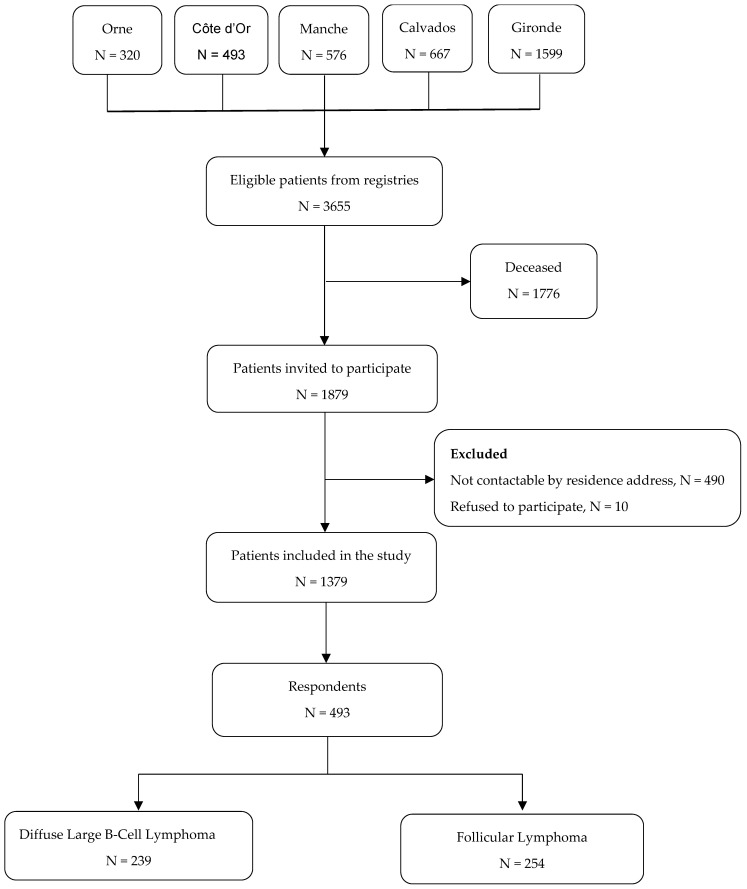
Flow chart of study.

**Figure 2 cancers-17-00711-f002:**
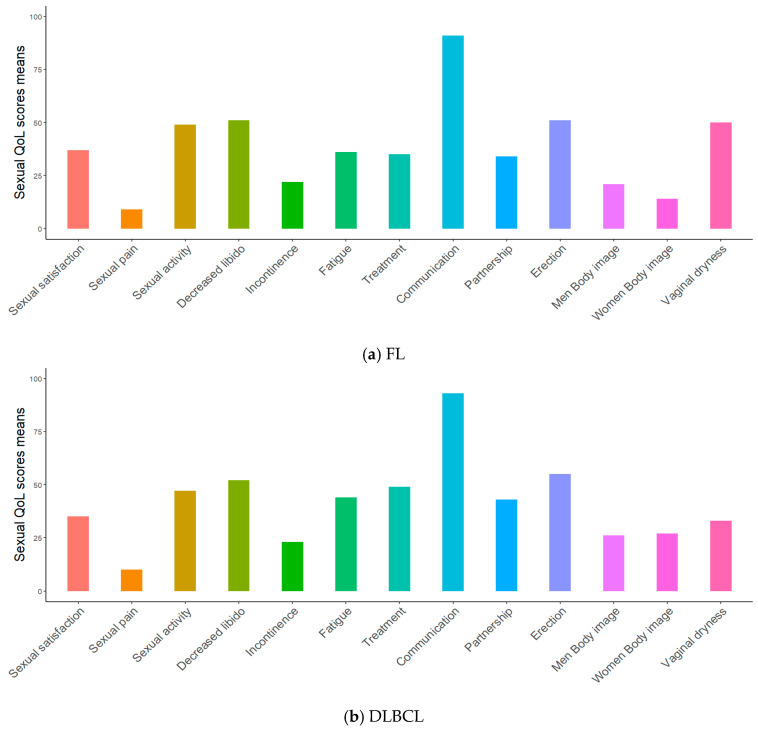
Subscale scores of sexual health in survivors of follicular lymphoma (**a**) and diffuse large B-cell lymphoma (**b**). High scores indicate a high level of symptomatology or concern.

**Figure 3 cancers-17-00711-f003:**
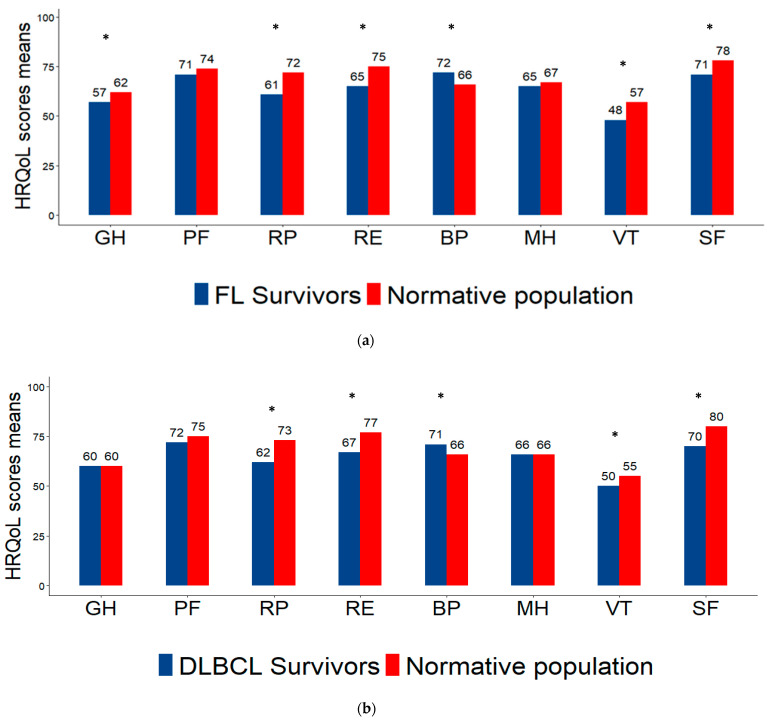
Subscale scores of the SF-12 questionnaire in the normative population matched for age and sex with survivors of follicular lymphoma (**a**) and diffuse large B-cell lymphoma (**b**). GH: general health; PF: physical functioning; RP: role limitations/physical health; RE: role limitations/emotional problems; BP: bodily pain; MH: mental health; VT: vitality; SF: social functioning. HRQoL: health-related quality of life. Higher scores indicate better HRQoL. * Significant differences at *p*-value < 0.05.

**Table 1 cancers-17-00711-t001:** Comparison of characteristics between non-respondents and respondents.

Characteristics	Non-Respondents N = 886	Respondents N = 493	Overall N = 1379	*p*-Value
Age at time of study (years)				0.7
Mean (SD)	65 (16)	65 (13)	65 (15)	
Median [IQR]	66 [55–76]	67 [58–75]	67 [56–76]	
Time since diagnosis (years)				0.007
Mean (SD)	8.6 (2.5)	8.2 (2.6)	8.5 (2.5)	
Median [IQR]	8.00 [7.00–11.00]	8.00 [6.00–10.00]	8.00 [6.00–10.50]	
Gender				0.088
Women	436 (49)	219 (44)	655 (47)	
Men	450 (51)	274 (56)	724 (53)	
NHL subtype				0.2
DLBCL	462 (52)	239 (48)	701 (51)	
FL	424 (48)	254 (52)	678 (49)	
Ann Arbor stage				0.12
I–II	320 (39)	155 (35)	475 (38)	
III–IV	497 (61)	291 (65)	788 (62)	
Missing	69	47	116	
R-CHOP chemotherapy				0.3
No	73 (9)	33 (7)	106 (8)	
Yes	777 (91)	439 (93)	1216 (92)	
Missing	36	21	57	
Administrative residence				<0.001
Calvados	135 (15)	74 (15)	209 (15)	
Côte d’Or	116 (13)	113 (23)	229 (17)	
Gironde	475 (54)	220 (45)	695 (50)	
Manche	104 (12)	63 (13)	167 (12)	
Orne	56 (6)	23 (4)	79 (6)	

R-CHOP: Rituximab-Cyclophosphamide-Hydroxy Doxorubicin-Vincristine-Prednisone. IQR: interquartile range; SD: standard deviation.

**Table 2 cancers-17-00711-t002:** Demographic, clinical, psychological, and socioeconomic characteristics of the study participants (N = 493).

Characteristics	Overall N = 493	DLBCL N = 239	FL N = 254	*p*-Value
Age at time of study (Years)				0.5
Mean (SD)	65 (13)	64 (16)	66 (11)	
Median [IQR]	67 [58–75]	67 [55–75]	67 [60–74]	
Time since diagnosis (Years)				0.13
Mean (SD)	8.29 (2.57)	8.12 (2.56)	8.46 (2.57)	
Median [IQR]	8.00 [6.00–10.00]	8.00 [6.00–10.00]	8.00 [6.00–10.75]	
Gender				0.4
Women	219 (44)	102 (43)	117 (46)	
Men	274 (56)	137 (57)	137 (54)	
Administrative residence				0.7
Calvados	74 (15)	40 (17)	34 (13)	
Côte d’Or	113 (23)	53 (22)	60 (24)	
Gironde	220 (45)	102 (43)	118 (46)	
Manche	63 (13)	31 (13)	32 (13)	
Orne	23 (4.7)	13 (5.4)	10 (3.9)	
Education level [1]				0.4
University or higher	172 (36)	88 (38)	84 (34)	
Secondary	304 (64)	142 (62)	162 (66)	
Alcohol consumption				0.8
≥1/month	264 (54)	128 (54)	136 (54)	
≥1/week	150 (31)	71 (30)	79 (32)	
No	73 (15)	38 (16)	35 (14)	
Smoking				0.07
≥11 cigarettes/day	19 (3.9)	6 (2.5)	13 (5.2)	
1–10 cigarettes/day	41 (8.4)	15 (6.3)	26 (10)	
No	426 (88)	216 (91)	210 (84)	
Ann Arbor stage				0.7
I–II	155 (35)	81 (36)	74 (34)	
III–IV	291 (65)	146 (64)	145 (66)	
BMI at time of study				0.9
Underweight	16 (3.5)	7 (3.2)	9 (3.9)	
Normal weight	194 (43)	91 (42)	103 (44)	
Pre-obesity	80 (18)	40 (18)	40 (17)	
Obesity	161 (36)	81 (37)	80 (34)	
Comorbidities				0.7
No	259 (53)	129 (54)	130 (52)	
Yes	227 (47)	109 (46)	118 (48)	
R-CHOP chemotherapy				<0.001
No	33 (7.0)	0 (0)	33 (14)	
Yes	439 (93)	237 (100)	202 (86)	
Monthly Income				0.5
EUR 0–1500	67 (14)	32 (14)	35 (14)	
EUR 1500–3000	203 (43)	101 (44)	102 (42)	
EUR 3000–5000	148 (31)	66 (29)	82 (34)	
≥EUR 5000	53 (11)	30 (13)	23 (9.5)	
Deprivation (EPICES Score)				>0.9
No	372 (77)	181 (77)	191 (77)	
Yes	112 (23)	55 (23)	57 (23)	
Anxiety				0.9
No	392 (82)	191 (82)	201 (82)	
Yes	86 (18)	41 (18)	45 (18)	
Depression				0.6
No	436 (91)	210 (91)	226 (92)	
Yes	42 (8.8)	22 (9.5)	20 (8.1)	
Self-esteem				0.8
Low	148 (31)	74 (32)	74 (31)	
High	226 (48)	107 (46)	119 (49)	
Moderate	101 (21)	52 (22)	49 (20)	
⁋ Social support availability				0.07
<13	218 (48)	96 (44)	122 (53)	
≥13	232 (52)	122 (56)	110 (47)	
⁋ Social support satisfaction				0.5
<30	196 (44)	91 (42)	105 (45)	
≥30	254 (56)	127 (58)	127 (55)	

R-CHOP: Rituximab-Cyclophosphamide-Hydroxy Doxorubicin-Vincristine-Prednisone. BMI, body mass index; IQR: interquartile range; SD: standard deviation. ⁋: Social support availability and satisfaction were categorized according to the median.

**Table 3 cancers-17-00711-t003:** Multivariable models to assess factors associated to HRQoL for each scale of the SF-12 questionnaire.

	GH		PF		RP		RE		BP		MH		VT		SF	
Variables	β	*p*	β	*p*	β	*p*	β	*p*	β	*p*	β	*p*	β	*p*	β	*p*
Age at time of study	−0.2	0.0001	−0.2	0.04	−0.3	0.009	−0.2	0.006	−0.3	0.004						
Time since diagnosis															1.2	0.009
Sexual satisfaction problem									−0.2	0.007	−0.2	0.001	−0.2	0.0006	−0.2	0.01
Gender		0.01								0.007						
Men	4.2								6.8							
Women	Ref								Ref							
BMI at time of study				0.008												
Underweight: <18.5			−5.4													
Pre-obesity: 25–30			−11.7													
Obesity: ≥30			−9.6													
Normal weight: 18.5–25			Ref													
Comorbidity		<0.0001		<0.0001		<0.0001		<0.0001		<0.0001		0.007		0.005		<0.0001
Yes	−9.7		−23.4		−19.4		−11.5		−16.2		−4.7		−6.9		−10.8	
No	Ref		Ref		Ref		Ref		Ref		Ref		Ref		Ref	
Income				0.005		0.0005		0.006		0.001		0.002				
€1500–3000			0.5		6.2		4.1		4.4		2.4					
€3000–5000			7.7		9.4		8.6		12.2		7.2					
≥€5000			16.3		19.6		13.9		15.7		12.5					
€0–1500			Ref		Ref		Ref		Ref		Ref					
EPICES deprivation		0.001		0.0006		0.006		0.0002						0.01		0.0004
Yes	−7		−13.4		−8.4		−10.7						−7.6		−10.7	
No	Ref		Ref		Ref		Ref						Ref		Ref	
Anxiety						0.01		<0.0001				<0.0001				<0.0001
≥11					−7.7		−13.9				−16.4				−17.5	
<11					Ref		Ref				Ref				Ref	
Depression		<0.0001				0.002		0.001		0.01		0.0001		0.001		
≥11	−13.9				−14.8		−14.5		−11.4		−13.1		−15.6			
<11	Ref				Ref		Ref		Ref		Ref		Ref			
Self-esteem								0.001				0.0001		0.01		
Low							−0.6				−4.6		2.3			
High							7.8				4.1		8.1			
Moderate							Ref				Ref		Ref			

GH: general health, PF: physical functioning, RP: role physical, RE: role emotional, BP: bodily pain, MH: mental health, VT: vitality, SF: social functioning.

## Data Availability

The data underlying this article will be shared on reasonable request to the corresponding author.

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
