# Peer review of "Health-Related Quality of Life in Long-Term Survivors of Non-Hodgkin Lymphoma: A French Population-Based Study"

_cancers, 2025, doi:10.3390/cancers17040711_

Round 1
Reviewer 1 Report
Comments and Suggestions for Authors
The authors have presented an evaluation of health-related quality of life (HRQoL) in long-term survivors of non-Hodgkin lymphoma (NHL) using a French population-based study. They have compared the HRQoL of 493 NHL survivors to the general French population and identifies key sociodemographic, clinical, and psychological factors influencing HRQoL. The findings reported here may provide insights for supportive care interventions in NHL survivors.
Recommended revisions:
1. The relatively low response rate (36%) may lead to potential selection bias that could in turn skew the results. The authors should address any key differences (or lack thereof) in available key variables between respondents and non-respondents.
2. The gender differences in HRQoL outcomes is interesting (males have better HRQoL). Have such gender related discrepancies in health outcomes been reported elsewhere, maybe in other studies on cancer survivors?
3. The ‘bodily pain’ metric being significantly lower in NHL survivors compared to the general population contradicts expectations. Are there any underlying biological mechanisms known or hypothesized that may explain this? Could this be due to bias in responses (i.e. altered perception of pain over time following chemo/radio therapy)?
4. Statistically significant associations between HRQoL and sociodemographic, clinical, and psychological factors have been reported, but considerations about potential confounding factors, such as treatment type (chemotherapy/radiotherapy), duration and intensity have not been discussed.
If data for these questions are not available, the authors should acknowledge this as a limitation of the current study
Comments on the Quality of English LanguageThe manuscript shifts between past and present tense, especially when describing results and methodology, which makes it somewhat awkward to read. Some terminology has been used interchangeably, such as respondents and participants. These should be addressed to improve clarity.
Some terminology used in the text can appear to be vague for most readers (e.g. response shift phenomenon). These should be briefly defined.
Author Response
The authors have presented an evaluation of health-related quality of life (HRQoL) in long-term survivors of non-Hodgkin lymphoma (NHL) using a French population-based study. They have compared the HRQoL of 493 NHL survivors to the general French population and identifies key sociodemographic, clinical, and psychological factors influencing HRQoL. The findings reported here may provide insights for supportive care interventions in NHL survivors.
Recommended revisions:
- The relatively low response rate (36%) may lead to potential selection bias that could in turn skew the results. The authors should address any key differences (or lack thereof) in available key variables between respondents and non-respondents.
Response 1: For significantly different variables (Time since diagnosis and Administrative residence), an adjustment was performed including these variables in all multivariate analyses.
- The gender differences in HRQoL outcomes is interesting (males have better HRQoL). Have such gender related discrepancies in health outcomes been reported elsewhere, maybe in other studies on cancer survivors?
Response 2: We showed in the discussion that, Hjermstad et al. found that women reported a lower quality of life (Hjermstad MJ, Fayers PM, Bjordal K, Kaasa S. Using reference data on quality of life - the importance of adjusting for age and gender, exemplified by the EORTC QLQ-C30 (+3). Eur J Cancer. 1998;34:1381–1389.)
- The ‘bodily pain’ metric being significantly lower in NHL survivors compared to the general population contradicts expectations. Are there any underlying biological mechanisms known or hypothesized that may explain this? Could this be due to bias in responses (i.e. altered perception of pain over time following chemo/radio therapy)?
Response 3: Our study showed that: The ‘bodily pain’ metric being significantly lower in NHL survivors compared to the general population.
Psychological mechanisms for example the response shift phenomenon may explain this. Indeed, the response shift phenomenon is a change in the meaning of one's self-evaluation of a target construct as a result of a change in the respondent's internal standards of measurement, a change in the respondent's values, or a redefinition of the target construct, in our study, the pain.
- Statistically significant associations between HRQoL and sociodemographic, clinical, and psychological factors have been reported, but considerations about potential confounding factors, such as treatment type (chemotherapy/radiotherapy), duration and intensity have not been discussed.
If data for these questions are not available, the authors should acknowledge this as a limitation of the current study
Response 4: The variable R-CHOP chemotherapy (No/Yes) was not statistically significant in the multivariable analysis. However, the duration and intensity of R-CHOP chemotherapy were not available. We have added these in the limits of the study.
Comments on the Quality of English Language
The manuscript shifts between past and present tense, especially when describing results and methodology, which makes it somewhat awkward to read. Some terminology has been used interchangeably, such as respondents and participants. These should be addressed to improve clarity.
Some terminology used in the text can appear to be vague for most readers (e.g. response shift phenomenon). These should be briefly defined.
Response: We revised tense in the manuscript and we choose “respondents” to refer at patients who returned QoL questionnaires. Moreover we defined the response shift phenomenon

Reviewer 2 Report
Comments and Suggestions for Authors
This manuscript is very interesting. It focuses on the Health-Related Quality of Life in survivors of non-Hodgkin lymphoma.
The authors adequately describe the background of the theme. However, there is a lack of information regarding the study design that they clarify just in the last paragraph of the discussion or as a limitation. The authors should include clearly from the beginning of the Study design description, not just a population-based study, but also a study that Will give information restricted to a cross-sectional approach no matter the prolective phase for data collection.
On the other hand, they were very self-indulgent about the success of the method employed for data collection; a response rate below 40% is a non-representative to generalize, and of course, they can not say that their study gave information as reference values for QoL for other studies, it Will be very valuable information but not necessarily representative for the whole population.
The paper is still of interest, and it may be modified if the authors consider it appropriate to extend the mentioned points.
Author Response
This manuscript is very interesting. It focuses on the Health-Related Quality of Life in survivors of non-Hodgkin lymphoma.
The authors adequately describe the background of the theme. However, there is a lack of information regarding the study design that they clarify just in the last paragraph of the discussion or as a limitation. The authors should include clearly from the beginning of the Study design description, not just a population-based study, but also a study that Will give information restricted to a cross-sectional approach no matter the prolective phase for data collection.
Response: We added information on the study design at the beginning of the Study.
On the other hand, they were very self-indulgent about the success of the method employed for data collection; a response rate below 40% is a non-representative to generalize, and of course, they can not say that their study gave information as reference values for QoL for other studies, it Will be very valuable information but not necessarily representative for the whole population.
Response: We revised the conclusion of our results regarding a response rate below 40%
The paper is still of interest, and it may be modified if the authors consider it appropriate to extend the mentioned points.
